# Classification of ECG signal using FFT based improved Alexnet classifier

**Arun Kumar M.** [1]*, **Arvind Chakrapani**[2]

**1** Department of ECE, Karpagam Academy of Higher Education, Coimbatore, India, **2** Department of ECE, Karpagam College of Engineering, Coimbatore, India

* arunkumar.m@kahedu.edu.in

**Data Availability Statement:** The database described in the following directory can be downloaded without charge from http://physionet.org/physiobank/database/mitdb/ (located at MIT, in Cambridge, MA, USA) and from PhysioNet mirrors

## Abstract

Electrocardiograms (ECG) are extensively used for the diagnosis of cardiac arrhythmias. This paper investigates the use of machine learning classification algorithms for ECG analysis and arrhythmia detection. This is a crucial component of a conventional electronic health system, and it frequently necessitates ECG signal reduction for long-term data storage and remote transmission. Signal processing methods must be used to extract the function of the morphological properties of the ECG signal changing with time, which is difficult to discern in the typical visual depiction of the ECG signal. In biomedical research, signal processing and data analysis are commonly employed methodologies. This work proposes the use of an ECG arrhythmia classification method based on Fast Fourier Transform (FFT) for feature extraction and an improved AlexNet classifier to distinguish the difference between four types of arrhythmia conditions that were collected from records. The Convolutional Neural Network (CNN) algorithm's results are compared to those of other algorithms, and the simulation results prove that the proposed technique is more effective for various parameters. The final results of the proposed system show that its ability to find deviations is 20% better than that of traditional systems.

## 1. Introduction

Cardiovascular disease is currently a severe hazard to life and health, and its incidence is increasing year after year. As a result, it is critical to concentrate on cardiovascular disease diagnosis and prevention. An electrocardiogram is the most popular non-invasive screening test to identify a specific cardiac issue. It keeps track of the heart's function throughout time. World Health Organization (WHO) statistics reveal that heart disease accounts for around one-third of all fatalities worldwide each year. Coronary heart disease has become one of the most common causes of death due to non-communicable and non-infectious conditions worldwide [1, 2]. Lifestyle, occupation, and diet are some of the leading causes of these disorders. Therefore, preventive and early diagnosis is the key to successful clinical treatment. The irregular function of the sinus node is the leading cause of cardiac insufficiency (SA node). The sinus node sends and receives electric impulses to control how the heart contracts and relaxes [3].

worldwide. All the datasets are granted to be used for research purpose without permission and consent.

**Funding:** The author(s) received no specific funding for this work.

**Competing interests:** The authors have declared that no competing interests exist.

Electrocardiogram (ECG) signals are generated due to these electrical pulses. As a result, monitoring the ECG signal can precisely represent the heart's bioelectrical activities. The heart's electrical activity is represented by an electrocardiogram, which is a temporary physiological signal. It has been utilized to find abnormal patterns in heartbeats and assess other factors, including heartbeat regularity and psychological stress [4, 5]. The electrocardiogram is also one of the most widely utilized approaches for detecting cardiac abnormalities. It gives information about arrhythmia and electrical activity that can be used to diagnose [6].

ECG machines are both safe and affordable. On the other hand, noise and other distortions might generate peaks in the ECG signal. The patient's body movements, body electrode movements, and power line disturbances are all examples of artifacts. Distortion and artifacts must be eliminated from the ECG signal to ensure accurate ECG analysis. Various transformations are performed to reduce noise and artifacts from ECG signals. The wavelet transform [7] is one of the most extensively used transformers. However, medical staff can not quickly diagnose the disease. Using only the appearance of the ECG signal is therefore not a correct way to detect any infection and a different function of these signals can help see disease [8, 9]. The following steps are used to classify ECG signals:

- ECG pre-processing

- PQRST wave reference point detection

- Function extraction and classification

The pretreatment step processes the ECG signal to remove artifacts added during signal collection. After removing the noise, the function of extracting the ECG signal is required. The FFT is an effective feature extraction method, that is used to identify these vertices. The feature vector generated in the feature extraction step must contain the minimum number of features for successful classification. The classification step consists of one or more classifiers that identify data categories specified by attribute vectors. Choosing a specific classifier can result in a higher classification rate than a specific cardiac variant. Arrhythmias are an essential group of cardiovascular diseases. An electrocardiogram is used to detect cardiac arrhythmias. The ECG is a vital modern medical device that registers the heart's excitability, conductivity, and recovery process. An ECG is a primary tool that doctors use to diagnose heart disease. ECG signals easily identify most heart diseases. However, human expertise is required to assess heart disease. For better analysis, CAD tools help clinicians with better treatment. This article proposes a convolutional neural network technique, AlexNet, based on the Fast Fourier Transform (FFT). This design classified the input ECG signal with a modified AlexNet neural network classifier.

The researchers also propose several other methods, but all have unavoidable drawbacks. Identifying the type of abnormality from detected ECG signal peaks is problematic because the two signals possess similar trends but may vary in disease. In other cases, two signals may exhibit different behaviors while exhibiting the same condition. As a result, it is critical to build an effective detection algorithm for identifying heart abnormalities. Since everyone's ECG signal has some unique pattern, it seems inappropriate to use a predefined mother wavelet in all cases. Using a predefined mother wavelet for all subjects may lead to approximations, which may cause misunderstanding of the ECG signal in the wrong circumstances [10]. Another drawback of conventional ECG classifiers is that they cannot provide personalized results. The intrinsic changes in ECG waveform shape across individuals due to gender, age, obesity index, genetic diversity, and other factors cannot be overlooked. Technology-based on artificial neural networks is applied for ECG signal classification. Comprehensive classifiers based on static ANNs perform poorly [11].

The Discrete Wavelet Transform (DWT) is the most commonly used function extraction algorithm, which divides an input signal into multiple levels. This degradation technique provides the entire ECG signal information. However, the received ECG signal's peak value from the DWT will be incorrect [12].

Researchers are using neural networks to solve challenges in medical diagnostics. This makes it possible to run "end-to-end" algorithms to predict real-time information on the fly, improve the efficiency of training methods, and quickly adapt to a broader range when large amounts of data are available.

Some of the major heartbeat problems investigated in the literature are Right Bundle Branch Block (RBBB), Premature Atrial Contraction (PAC), and Premature Ventricular Contraction (PVC). These four types are simultaneously presented to improve the AlexNet classifier. As a result, a deep learning system is presented that can distinguish different sorts of anomalies depending on the patient's condition. A deep learning-based protocol identifies the patient's susceptibility to the disease (more severe, standard) in the suggested treatment.

The organization of the research work is as follows: Section 2 describes the literature survey of heart disease prediction. Section 3 details the proposed methodology. Section 4 demonstrates the results and analysis of the research. The research is summarized in the conclusion.

## 2. Literature review

A deep time-frequency representation and progressive decision fusion for ECG classification. using a new deep learning convolutional neural network-based ECG signal classification method is proposed. A short-time Fourier transform converts the ECG signal into the time-frequency domain. ECG data of various durations train scale-specific deep convolutional neural networks. Finally, a strategy for fusing the decisions of a specific scale model into a more precise and stable model is provided using an advanced online decision fusion method [13].

An Ensemble ECG classification classifier using expert features and deep neural networks is described. This document proposes to ENCASE, which combines expert functions and a Deep Neural Network (DNN) for ECG classification. ENCASE is a flexible framework that supports incremental feature extraction and classification updates. Experiments have shown that ENCASE is superior to other methods. An investigation of four classes of ECG data classification reported an F1 score of 0.84 [14]. ECG biometrics using spectrograms and deep neural networks is created to leverage the latest development in biometrics systems based on ECG spectral procedures and CNNs. This article will briefly introduce ECG biometrics and then present the newest biometrics. One of the advantages of this algorithm is that it is robust against momentary fluctuations in signal acquisition because it can correctly identify spectrograms with short time offsets. The results obtained are good, but there is still potential for improvement [15].

Learning and synthesizing biological signals using deep neural networks is proposed. New algorithms for signal reconstruction and source detection with very noisy data in biomedical engineering are explained. Each signal is preprocessed, divided, and quantized into a specified number of classes matching the sample size and then delivered to a model that includes an embedded matrix, three GRU blocks, and a softmax function. By considering the initial value, the upcoming conceptual value is obtained through the modifications in the internal parameters of the network. Random values are generated, and analog signals are made by reentering the values in the network. It has been demonstrated that this synthetic process may be used to describe signals from various physiological sources [16].

A convolutional recurrent neural network for ECG classification is developed. This article proposes two deep neural network algorithms for classifying electrocardiogram recordings,

regardless of length. The first schema is a deep CNN with method-based time-consuming feature aggregation. The second architecture combines function extraction convolutional layers and long-term memory layers for the temporal collection of functions. The dual architecture proved to be better than the first; achieving an F1 score of 82.1% on the hidden challenge test set [17].

An ECG cardiac arrhythmia classification using extended signals in time series and in-depth learning methods is described. This data set contained ECG samples from 47 subjects, initially recorded at a sampling rate of 360 Hz and then sampled to 125 Hz. This article proposes a pre-treatment technique that significantly improves the accuracy of deep learning models for ECG classification and improves training stability through an improved deep learning architecture. The system can achieve more than 99% accuracy using this preprocessing technique and deep learning model without overfitting the model [18].

A patient-specific ECG classification using an integrated long-term memory and convolutional neural network is elaborated. Long-term memory and convolutional neural networks are combined in this paper to create an automated patient-specific ECG categorization technique. From steady heartbeats, LSTM extracts temporal data like Heart Rate Variability (HRV) and correlations between heartbeats, whereas CNN captures specific morphological properties of the current heartbeat. In addition, novel clustering algorithms have been developed to identify the most representative patterns from regular training data. SVEB sensitivity and positive prediction frequency rose by 8.2% and 8.8% or greater, respectively, compared to earlier research [19].

ECG biometrics using wavelet analysis combined with stochastic randomized forest reveals a new algorithm that improves the accuracy and resilience of human biometric identification by using ECGs from mobile devices. This algorithm combines the benefits of benchmarking and non-benchmark ECG capability, combining wavelet analysis with stochastic random forest machine learning to provide a fully automated two-level cascade classification system. These findings confirm the suggested biometric algorithms' accuracy and effectiveness and their utility in applications like telemedicine and cloud data security [20].

A novel function is proposed to extract ECG signal classification and fast Fourier transform for neural networks. This research describes a new approach for classifying complicated cardiac disorders based on ECG data. R peak identification and pulse extraction use signal filtering and rapid Fourier techniques, followed by neural network-based signal modeling and categorization of ECG data. The MLP demonstrates good classification performance using the same recorded test samples as the training mode [21].

In [22] ECG signal classification using deep learning techniques based on the PTB-XL dataset is developed. The research work aims to build a deep neural network that can automatically classify necessary ECG signals. Data from the PTB-XL database is used in the survey. The first is based on folding networks, the second on SincNet, and the third on folding webs with entropy-based functions added. Correspondingly, training sets, validation sets, and test sets make up 70%, 15%, and 15% of the data set.

A review of ECG arrhythmia classification using a deep neural network is explained in [23]. This paper describes a new DL approach for categorizing ECG signals. ResNet, InceptionV3, Gated Recurrent Unit (GRU), and Long Short-Term Memory are some of the DL approaches in this work. LSTM and CNNs are most often used to extract valuable characteristics.

In [24] new feature extraction is created for ECG signals for early detection of heart arrhythmia. The main properties of the ECG signals P, Q, R, S, and T and their segments and distances are discussed in this article. To extract the desired properties from the ECG signal, use the Walsh-Hadamard Transform (WHT) and the Fast Fourier transform (FFT). These results were produced using Matlab, and the derived functions were then applied to patient

records to detect cardiac arrhythmias. The generated Excel file can be used to classify and detect various irregularities.

Feature Extraction of Heart Signals using Fast Fourier Transform is proposed in [25]. This study aimed to categorize cardiac signals or data from Physiobank, the MIT-BIH Arrhythmia Database, and the MIT-BIH Normal Sinus Rhythm Database. Using the Fast Fourier Transform function extraction approach, process the data. Before being employed in the classification procedure, the outcomes of the function extraction approach were chosen. A backpropagation neural network is used for classification. According to the study, the function extraction approach of the Fast Fourier Transform provided an 87% classification accuracy by extracting 64 data points for classification following the FFT procedure and backpropagation.

In [26–32], SpEC based on Stockwell Transform (ST) and 2D Residual Network (2D-ResNet) is proposed to improve ECG beat classification techniques with a limited amount of training data. ST is used to represent ECG signals in the time-frequency domain and provides frequency-invariant amplitude response and dynamic resolution. The generated ST images were used as input for the proposed 2D-ResNet and the five ECG beats were classified in a patient-specific manner, as recommended by the Association for the Advancement of Medical Devices (AAMI).

## 2.1 Problem statement

In the literature, there are numerous interpretations of ECG beat classification using a variety of techniques, including Artificial Neural Networks (ANN), Self-Organizing Maps (SOM), Support Vector Machine (SVM) classifiers, Soft Independent Modeling of Class Analogy (SIMCA), deep learning, Complex Support Vector Machines (CSVM), decision trees, and Convolution Neural Network (CNN).

SVM exhibits poor behavior in class instabilities, but methods of handling have been developed, including using hierarchical SVM or SVM weighted by each class. The drawback of ANN is that, in complex problems, it may not always be possible to find an optimization, and the training algorithm is not guaranteed to achieve a global optimization.

Due to the high computational cost during the test phase, the k-Nearest Neighbor (kNN) method has limited application in real-time scenarios. The decision tree is not commonly used because it can only handle a limited number of features and the rule-based approach performs the worst.

## 2.2 Major contributions

Numerous algorithms have been used in research, including random forests and decision tree ensembles as well as the non-linear classifier Support Vector Machine (SVM). Manual feature extraction is necessary for this algorithm. Researchers use neural networks to address this issue to advance not only medical diagnostics but also other fields of study. This increases the effectiveness of training techniques, makes it possible for the algorithm to be used "end-to-end," and makes it simpler for a wider range of people if large datasets are available. able to be modified.

The goal of this research is to develop a deep learning-based method for automatically detecting arrhythmias without the use of manual feature identification.

The suggested research has three stages:

- Noise reduction

- Feature classification using FFT

- Anomaly analysis using deep learning techniques

FFT is used to convert the time domain signal to frequency domain ECG signal for more accurate peak extraction. The results are then forwarded to the taxon, who will look for ECG abnormalities.

## 3. Proposed methodology

This article proposes improved AlexNet, a convolutional neural network technology based on Fast Fourier Transform (FFT). It extracts a more straightforward set of functions from the input ECG data. This design classified the input ECG signal using AlexNet's neural network classifier. Perform a Fast Fourier Transform (FFT) analysis for identification. ECG signal processing can also be performed using wave transformation techniques to detect RR intervals, QRS complexes, T-waves, and P-waves as shown in Fig 1. The signal is first preprocessed to eliminate noise. It then extracts the functions and implements on deep learning-based detection algorithm. The terminology utilized in the proposed methodology is detailed next.

### 3.1 ECG theory

An ECG is used to interpret the electrical impulse of the human heart. It varies from person to person, depending on the condition of the heart. Electrodes are placed on the skin's surface further to record the heart's electrical activity over time. ECG signals are non-standing waves [24]. An ECG beat segment is generated using python is shown in Fig 2.

### 3.2 Datasets

The MIT-BIH (Massachusetts Institute of Technology-Beth Israel Hospital) arrhythmia database [22] is used in the suggested technique. The database contains 48 records from 47 people. Each recording contains two channels (MLII and V5) of ECG signals for 30 ECGs chosen from a 24-hour recording. The Continuous ECG Signal Pass Band Filter uses a 0.1–100 Hz band pass filter to filter the signal and convert it to digital data. There is also an annotation file

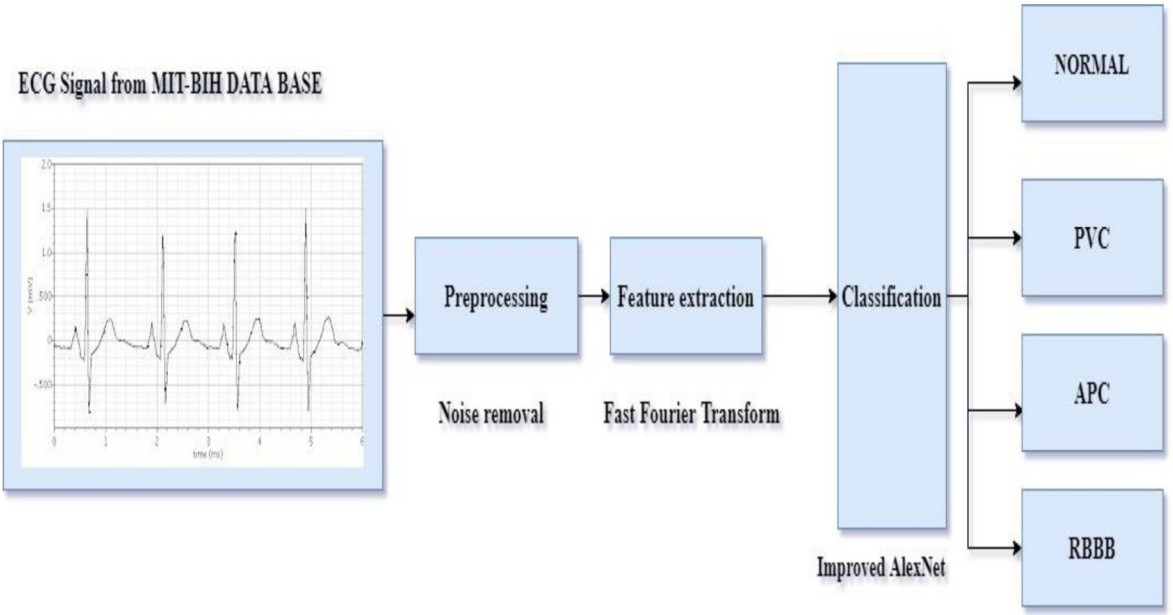

**Fig 1. ECG signal processing procedure.**

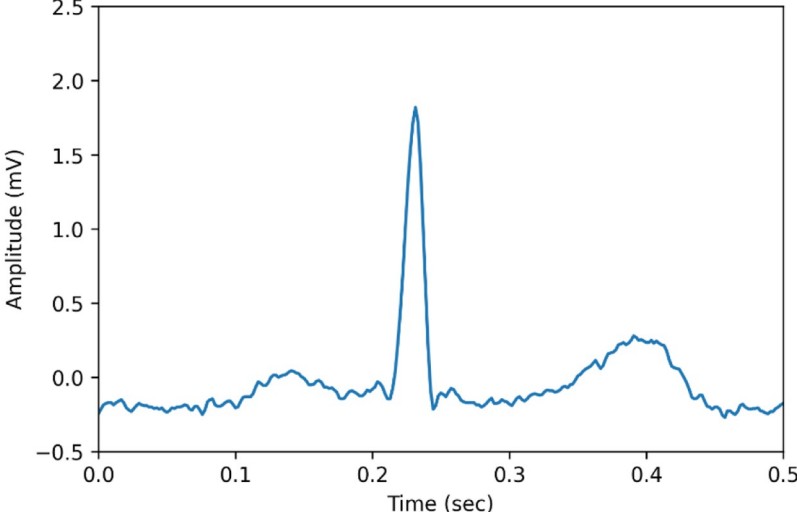

**Fig 2. ECG beat segment.**

for each record in this database. The annotation file contains information such as heartbeat occurrence time (R peak position) and heartbeat class. A heartbeat can be detected using 100 samples around the R peak. The database excludes four records with rhythmic beats and uses the remaining 44 records. A representative sample of clinical records for routines used as a general training set can be found in the first 20 records, which are numbered from 100 to 124. The final 24 records (numbers 200–234) featured abnormal heartbeats like ventricular and supraventricular arrhythmias. Use these records as a test set. The database described in the following directory can be downloaded without charge from http://physionet.org/physiobank/database/mitdb/ (located at MIT, in Cambridge, MA, USA) and from PhysioNet mirrors worldwide. All the datasets are granted to be used for research purposes without permission and consent.

**3.2.1 Right Bundle Branch Block (RBBB).** A normal management system interruption called a bundle branch block causes an abnormal QRS complex. The right branch block typically depolarizes the Right Ventricle (RV). In RBBB, there is no activation of the right branch block. Instead, a pulse is sent from the left ventricle to the right ventricle via the left ventricle (LV), depolarizing it.

**3.2.2 Premature Atrial Contraction (PAC) and Premature Ventricular Contraction (PVC).** When the heart's regular rhythm is disrupted by a premature or early beat, PAC and PVC occur. A PAC is a premature beat that originates in the atria. It is referred to as PVC if it arises from the ventricles.

## 3.3 Pre-processing

ECG data obtained from the database is much less noisy (taken directly from the patient). Still, externally induced high and low-frequency sounds such as DC tones, muscle contractions, breathing movements, electrode placement, etc. There are some familiar voices, such as voices from equipment. Therefore, a signal preprocessing step is required to remove noise in ECG recordings. Remove the average of 500 samples from each sample obtained by ECG to avoid unwanted noise signals in the entering ECG waveform. The signal's baseline amplitude is reduced to zero due to this action. The filter is tuned to allow low-frequency impulses while attenuating the high frequencies to minimize the noise of the high-frequency components.

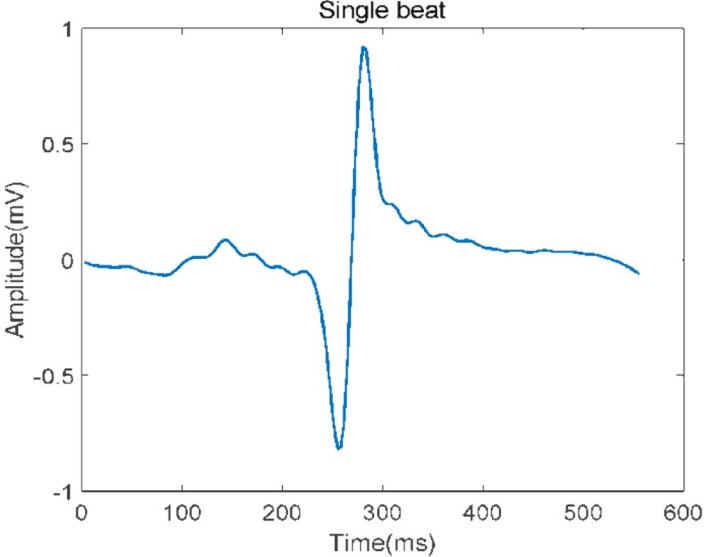

**Fig 3. Single heartbeat after denoising.**

The signal's baseline amplitude is reduced to zero due to this action. The filter is tuned to allow low-frequency impulses while attenuating the high frequencies contained in the erratic ECG signal to minimize the noise of the high-frequency components. The high pass filter allows high frequencies while attenuating low frequencies to minimize low-frequency noise as given in Fig 3.

ECG signal behavior is subject to several parameters, including the health, patient's age, and atmosphere. The Electro gram signal is measured from the patient's body, and the system adds noise to the signal throughout the recording process. Under different settings, the amplitude and value of the ECG signal vary from patient to patient as shown in Fig 4. As a result, a method for eliminating noise from ECG readings must be developed. ECG signal noise is caused by motion artifacts, power line failure of the signal, baseline manipulation, and attenuation losses. Various hardware design solutions can be used to reduce noise, such as power line interference and motion artifacts. After the noise has been removed, it is necessary to extract the properties of the ECG signal. The raw input of the ECG signal is prone to noise at the output due to the potential generated by the heart, resulting in attenuation losses. Therefore,

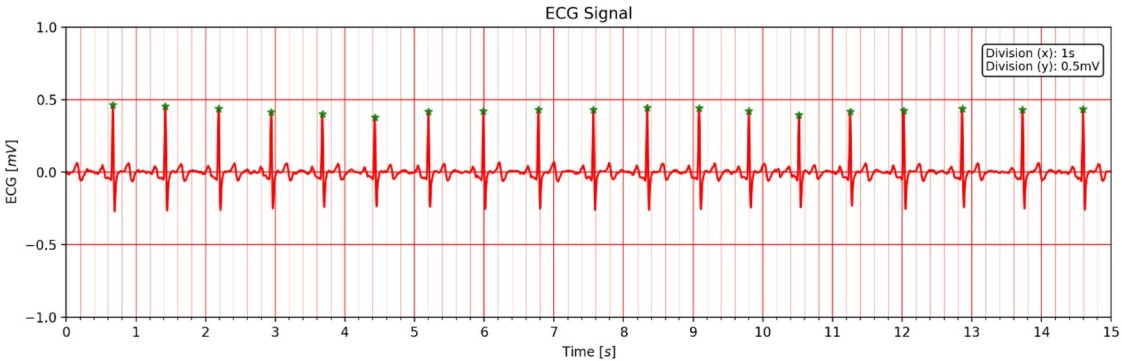

**Fig 4. ECG signal after median filtering.**

denoising is crucial to predict anomalies more accurately. Denoising of the ECG signal is performed using a relaxed median filter.

## 3.4 Feature extraction using fast fourier transform

A transform is a mathematical tool that moves from the time domain to the frequency domain. The transformation changes the representation of the signal by projecting the signal onto a set of essential functions but does not change the information content of the signal. Various types of feature extraction methods have been available for decades, including FFT [4], DFT [6], Short-Time Fourier transform (STFT) [13], and wavelet-based features of [3], features based on crossed wavelets. Handcrafted features are used to input traditional state-of-the-art classifiers such as SVM, Least Squares SVM, LIB-SVM, PNN, and LVQ. This article proposes an efficient data-independent technology Fast Fourier Transform coupled with AlexNet. Feature extraction algorithms can limit the number of reference points in the ECG signal by adding an effective threshold to the peak points. These peak points are found using the Fast Fourier Transform, an efficient feature extraction technique that includes numerous additional issues in addition to the PQRST signal.

Each complex ECG signal has a real and an imaginary component. The Fast Fourier Transform removes low frequencies from the ECG signal. The inverse fast Fourier transform is used to remove noise. The Fast Fourier Transform is used to transform the input signal from the dataset after it has been preprocessed by removing nulls. The extracted features were suitable for detecting arrhythmia in patient records, and the results were obtained using Matlab. The created Excel file can then be used to classify and detect various anomalies.

A periodic extension of the period [21, 33] can be used to derive the piece-wise continuous function F(t) defined in the interval t∈| 0, α |.

$$F(t) = \sum_{k=-\infty}^{\infty} C_k e^{i2\pi k \frac{t}{a}} \tag{1}$$

Function F(t) can be sampled at a discrete-time

$$t_j = j\frac{a}{N}, \quad j = 0, \ldots\ldots N$$

$$F_j = F(t_j) = \sum_{k=-\frac{N}{2}}^{\frac{N}{2}} C_k e^{i2\pi k j \frac{1}{N}}, \quad j = 0, \ldots, N \tag{2}$$

This extension has N+ 1 value $F_j$ and therefore N+ 1 coefficient $C_k$ can be calculated.

**3.4.1 QRS complex identification.**   First, the ECG signal is pretreated to remove power line noise and high-frequency interference. The ECG signal's Q, R, and S deflections are then identified, and the QRS complex is deduced from these deflections. This is a critical function for detecting arrhythmias. The complex QRS identification system works in three steps. PhysicalNet was used to collect ECG signals from the MIT-BIH arrhythmia database. The database's ECG signals are pre-processed to remove noise from power lines and high-frequency interference. The obtained data is then subjected to deflection identification.

**3.4.2 R peak detection.**   The first step is to extract relevant measurements from the target signal. Before extracting the ECG signal, the Q, R, and S deflections for each stroke were calculated. This is accomplished using an algorithmic script and the following method: The first goal is to detect R peaks as they appear. Simple Q and S scores can be used to detect Because of the QRS complex's uniqueness and the characteristic function of the R-peak, it can be easily

identified even with the most distorted ECG measurements. As a result, it is used to determine ECG function. To detect deflections, a method based on digital signal processing is used. First, the FFT is applied to the ECG signal in Eq 3.

$$X_k = \sum_{n=0}^{N-1} xne^{\frac{2\pi i}{N}nk}, \quad k = 0, \ldots . N-1 \tag{3}$$

Eq 4 is used to apply the inverse FFT to the resultant signal.

The signal is now filtered to detect the R peaks.

$$X_k = \frac{1}{N}\sum_{n=0}^{N-1} xne^{\frac{2\pi i}{N}nk}. \quad k = 0, \ldots . . N-1 \tag{4}$$

The signal obtained after the first pass is passed through the filter again after the second pass.

**3.4.3 Q peak detection.** The accuracy of the R points calculated above is adequate. A negative wave at the beginning of a QRS complex is referred to as a Q wave, and the Q point is the valley's minimum. As a result, to locate the Q point, it is positioned as a local minimum in a brief window (of about 0.05 seconds) surrounding the left side of the R point calculated in Eq 5.

$$\frac{dy}{dx} < 0 \forall x < x(t) \ and \ \frac{dy}{dx} > 0 \forall \ x > x(t) \tag{5}$$

**3.4.4 S peak detection.** After the R point found in the formula, the S point is first roughly defined as the location where the slope exhibits the first negative zero to positive zero crossing.

$$\frac{dy}{dx} < 0 \forall x < x(s) \ and \ \frac{dy}{dx} > 0 \forall \ x > x(s)$$

**3.4.5 RR intervals.** The deflection positional information is used to generate metrics for the RR interval, which is a medical indicator of ventricular heart rate. Two R peaks in consecutive beats are calculated to determine the RR interval, and their difference is computed. The heart beats per minute are 60/RR interval.

$$\text{Rate} = \frac{60}{RR \ interval} \ \ beats \ per \ minute$$

Multiple Cardiovascular Arrhythmias are detected using these features.

## 3.5 Improved AlexNet

AlexNet, a pre-trained deep CNN, was used to classify ECG signals. AlexNet is trained on millions of images to classify 1000 objects. The model consists of three fully linked layers and five convolutional layers. Three fully linked layers and five convolutional layers make up the model. The first AlexNet layer takes a filtered image with dimensions of 227 × 227 × 3, width, height, and depth (red, green, blue). The AlexNet architecture comprises 1000 connected layers, and the remaining layers are used for feature extraction [23, 34].

For each input image, AlexNet can produce a 4096-dimensional feature vector, such as by activating a hidden layer before the output layer. With 650,000 neurons and 60 million parameters, AlexNet is a massive structure. By preserving dropout and data expansion, the model effectively reduces the problem of overfitting. The CNN AlexNet was chosen for this study

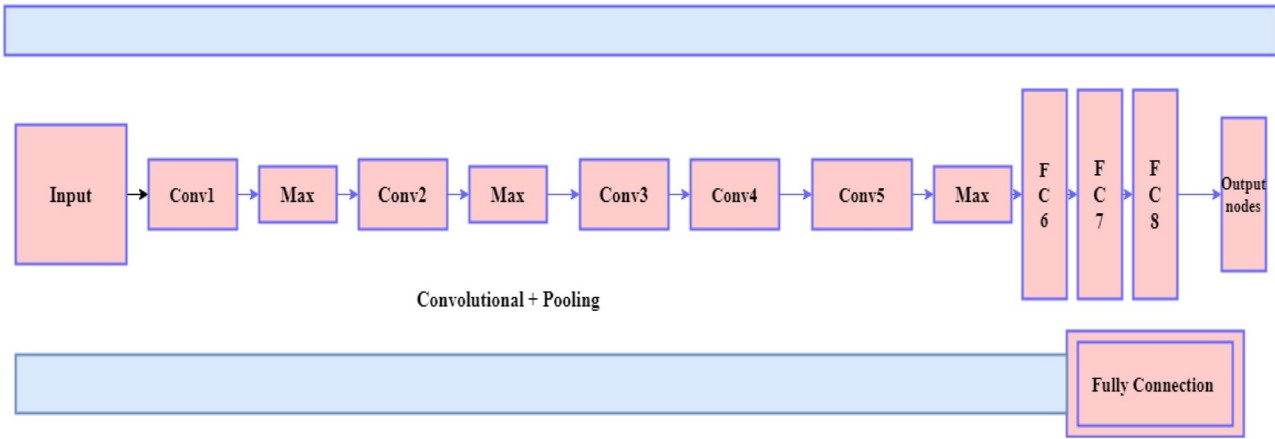

**Fig 5. Architecture of the AlexNet model.**

because it is the most commonly explored and provides an excellent balance of speed and accuracy. The AlexNet architecture is depicted in Fig 5.

The following characteristics distinguish the improvements proposed in this paper from the traditional AlexNet network classification algorithm:

- An additional convolution layer is introduced to the original AlexNet structure and the max-avg pooling technique is used to preserve the local receptive fields. This will provide more accurate image feature information.

- The Global Average Pooling (GAP) layer is substituted for the original fully connected FC layer, which significantly reduces the over-fitting effect without affecting the final features. The final result is unaffected in the absence of numerous calculations of network parameters, increasing network speed.

- The LRN layer is added to the convolution layer to avoid some unnecessary numerical issues; this effectively avoids neuron saturation. The BN layer in the proposed method is used after the convolution of each layer.

The AlexNet model has a large folded kernel. The step of the first folding layer limits image classification, resulting in a rapid drop in the resolution of the functional map and over-compressed spatial information. This document proposes an improved AlexNet model based on the design principles of Convolutional Neural Networks (CNN). The large convolution kernel is decomposed into a structural cascade of two small convolution kernels with a reduced number of steps. After the first layer, an additional folding layer is added to improve the low-level function or the spatial information integration process. The asymmetric folding core applies to the last three folding layers. Experiments with the two data sets show that the improved AlexNet model rating accuracy is higher than the AlexNet model rating accuracy. The improved AlexNet architecture is depicted in Fig 6.

Each ECG image is transmitted to the improved AlexNet in this classification stage. The individual ECG beats that were recovered from the Fourier coefficients were further divided into two groups for training and testing to facilitate classification. Using the FFT coefficient as the function vector in the classifier's input vector, the improved AlexNet classifier is used to differentiate between the four different types of ECG arrhythmias.

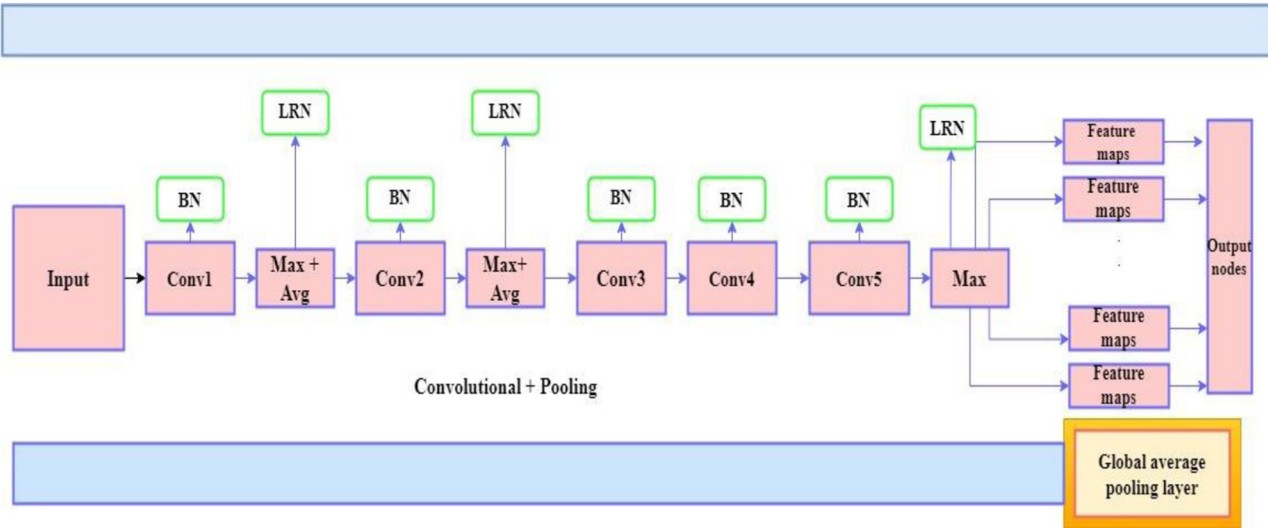

**Fig 6. Architecture of the improved AlexNet model.**

### 3.6 Transfer learning

This paper proposes an improved AlexNet using the Fast Fourier Transform (FFT). At first, ECG signal features are extracted using an efficient FFT. Then, anomalies in heart disease patients are classified using the proposed multipurpose genetic algorithm. AlexNet shows excellent classification efficiency, but training takes time. Fig 7 shows a basic schematic of transfer learning with AlexNet. Transferring previously acquired knowledge to a new model for in-depth learning without having to start over from the beginning is known as transfer learning.

As a result, the remaining layers are only initialized. After then, the structure is divided into two networks: a training network and a forwarding network. Pre-trained network parameters are trained for millions of images on ImageNet, and the extracted functions are always categorized. These parameters only need to be adjusted slightly based on the new input image. These

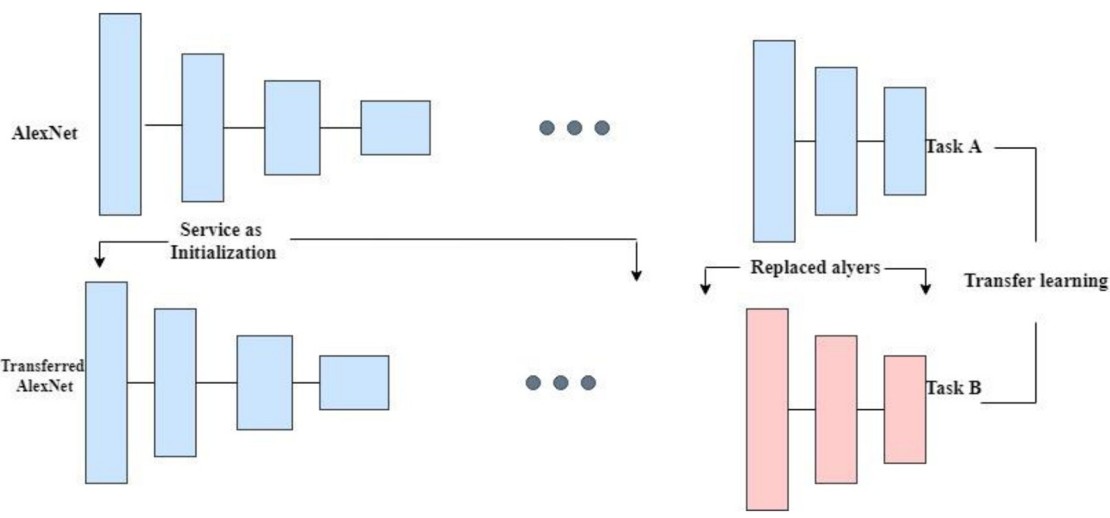

**Fig 7. The transfer learning process of AlexNet.**

parameters have little impact on the overall CNN training and are ideal for training entirely new classes of data sets.

The detection of anomalies and the activation of the patient's condition are classified. The fitness function of the multipurpose genetic algorithm is built after initialization, utilizing the convergence range of a specific anomaly. It is abnormal if the fitness value is less than the convergence range. The bias stage is then determined using a fuzzy-based technique in the second step. This stage indicates the output's degree of divergence. As a result, the type of anomaly and its status can be predicted. Fig 8 shows the overall contribution proposed research work.

## 4. Results and discussion

All experiments were performed using the Matlab R2021b programming environment. The performance of improved AlexNet was evaluated using an ECG dataset containing 1200 signal segments to classify various arrhythmias. 80% of the data in these 1200 records are used for training, and the remaining 20% is used for testing. Standard metrics evaluation: Accuracy (ACC), Sensitivity (ST), Specificity (SP), and Precision of Analyzing eight-layer Alexnet Model Function. Fig 9 shows the recorded ECG signal with and without noise.

The optimized values for early learning rate ($\eta$), mini-batch size, and the number of training iterations are 0.0002, 128, and 120, respectively, as shown in Fig 10. F (n) usually rises in proportion to the size of the box. The linear relationship between the double log plots indicates that there is scaling. That is, F (n) = n ^ $\alpha$. In this case, the variability can be characterized by a proportional exponent $\alpha$, where log F(n) is the slope of the line associated with log (n). A $\alpha$ of 0.5 corresponds to white noise, $\alpha$ = 1 corresponds to l / f noise, and $\alpha$ = 1.5 corresponds to brown noise or random walk. A good linear fit from log F(n) to log (n) plot (DFA plot) shows that F (n) is proportional to n as obtained in Fig 11.

The starting learning rate is 0.0002, the mini-batch size is 128, the number of iterations is 120, and the classifier's detection accuracy is 99.7% when using raw ECG data. As illustrated in Fig 12, these two forms of confusion matrices correlate to the result's accuracy, sensitivity, specificity, anomaly prediction (Precision), and mean prediction utilizing the proposed AlexNet classifier, respectively. The accuracy and loss curves are shown in Figs 13–15 as a function of the initial learning rate ($\eta$), mini-batch size, and the number of iterations, respectively.

Normal, Premature Ventricular Contraction (PVC), Premature Atrial Contraction (APC), and Right Bundle Branch Block (RBBB) strokes were all collected from this database and chosen for this study. Several varieties that display characteristics simultaneously are selected from the four types listed above. These findings match recordings from the precordial and limb leads and patients I17, I20, I22, and I71. From this database, a total of 1200 heartbeats were recovered. These beats are used in AlexNet Classifier's classification training and performance evaluation.

The ROC curves of the model are shown in Fig 16.

Table 1 displays the test set specificity, sensitivity, and positive prediction performance of the improved AlexNet Classifier with a real Gaussian core. It displays the percentage of correct classifications in terms of ST, SP, PP, and ACC for a particular category (stroke type). According to simulation data, the RBBB type has the highest accuracy of 98.33% in each class, while the PVC type has the lowest accuracy of 96.50%. The classification accuracy was 97.17% for APC types and 97% for NORMAL kinds for the other categories. Furthermore, the RBBB types' classification sensitivity was 99.85%. For all four categories, the classification specificity was greater than 94%.

FFT is used to detect peak amplitude and efficiently classify beats. It is discovered that the suggested FFT is 99.7% efficient in peak detection when compared to existing discrete wavelet

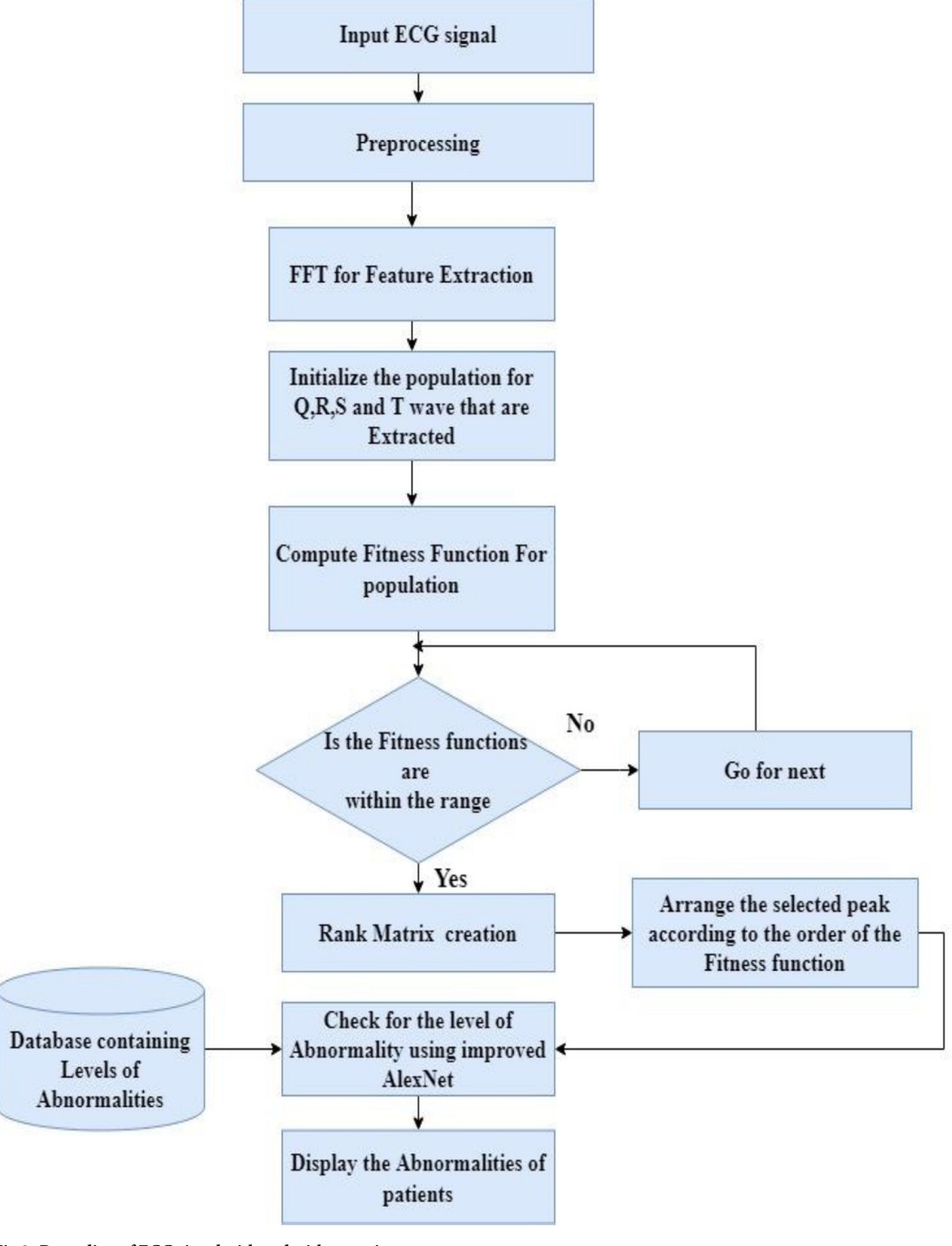

**Fig 8. Recording of ECG signal with and without noise.**

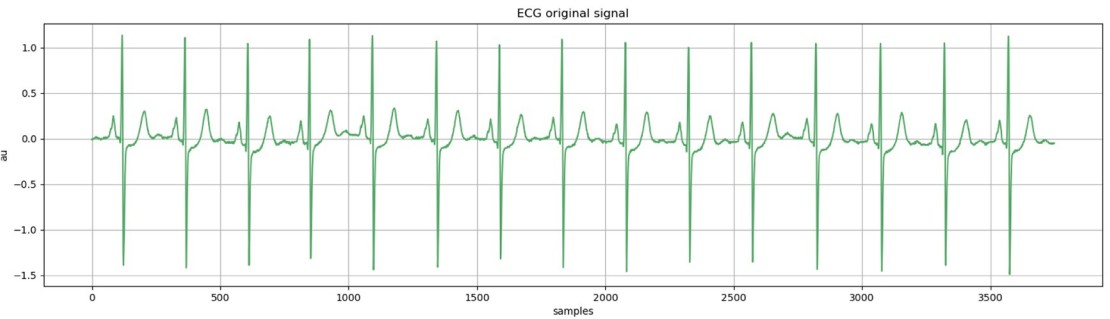

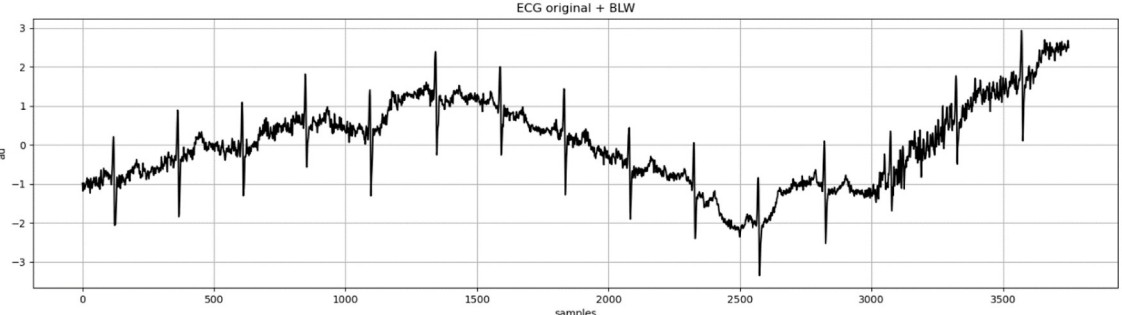

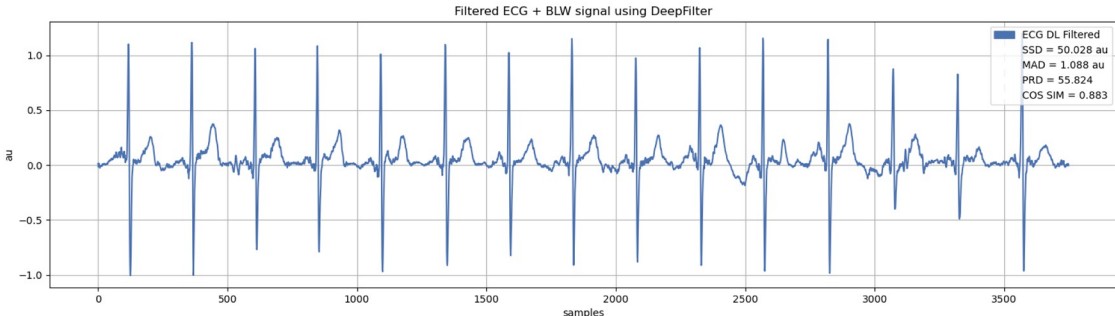

**Fig 9. Flow diagram of the improved AlexNet model.**

transform techniques.

$$Accuracy = \frac{TP + TN}{TP + TN + FP + FN} \tag{6}$$

$$Specificity = \frac{TN}{TN + FP} \tag{7}$$

$$Sensitivity = \frac{TP}{TP + FN} \tag{8}$$

$$Precision = \frac{TP}{TP + FP} \tag{9}$$

Where,

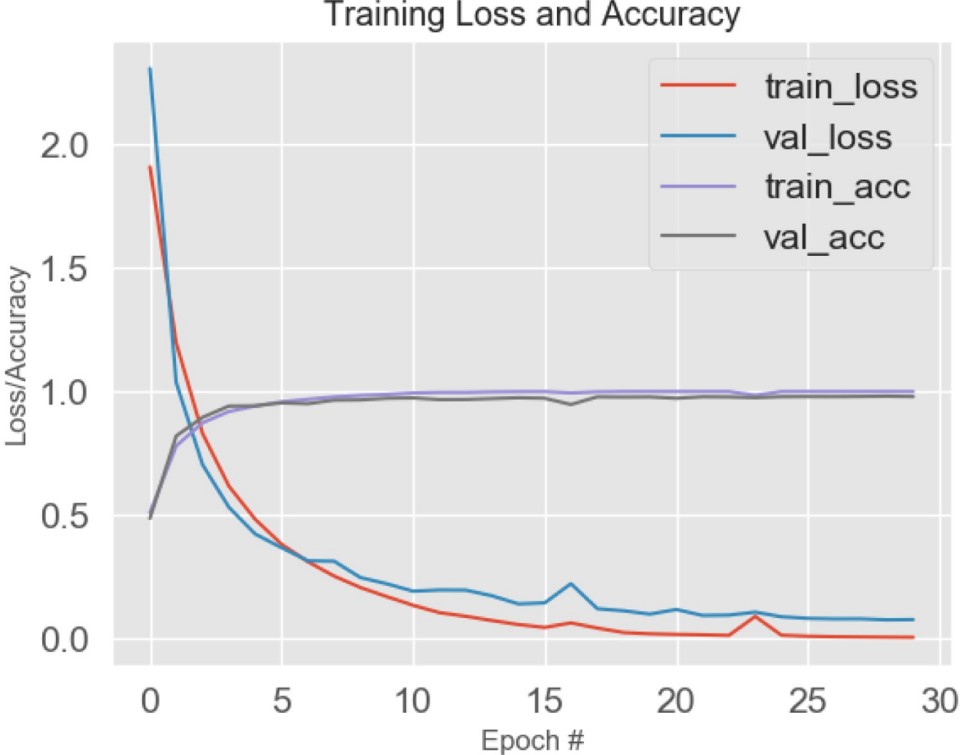

**Fig 10. Training and validation performances using a proposed model with ECG datasets.**

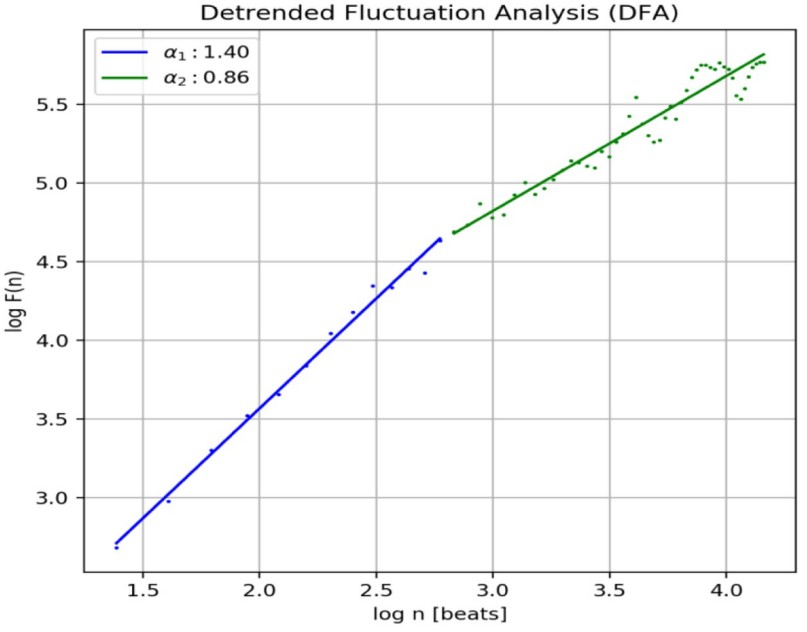

**Fig 11. Detrended Fluctuation Analysis (DFA).**

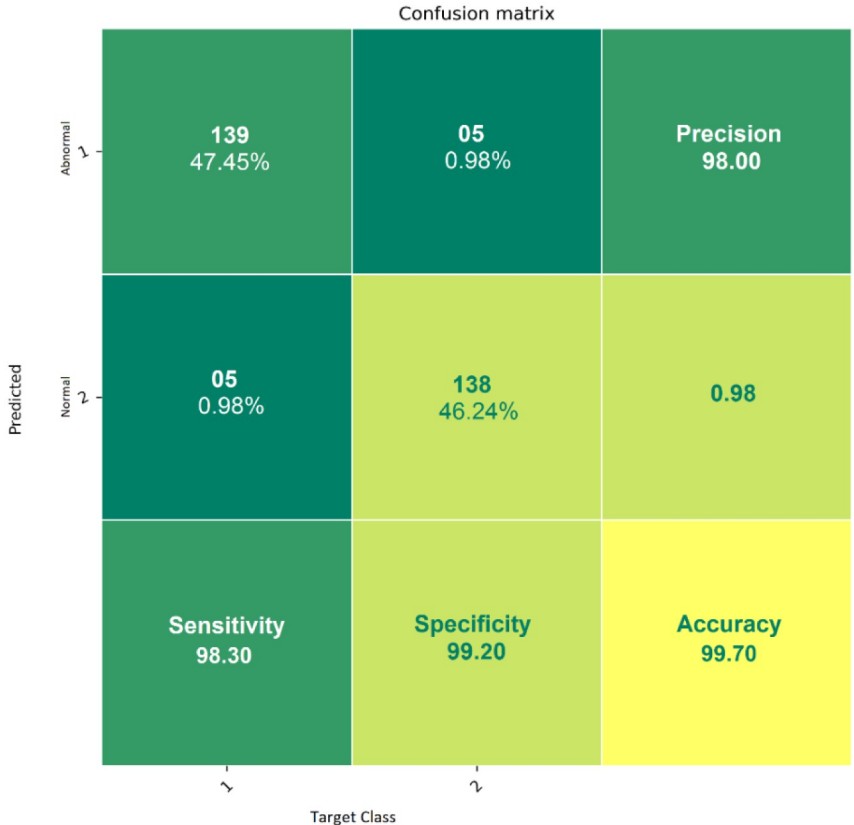

**Fig 12. The proposed deep learning confusion matrix model based on AlexNet.**

TP (True Positives)—Total number of heart sounds correctly classified as abnormal.
TN (True Negatives)—the total number of heart sounds correctly classified as normal
FP (False Positives)—False-positive (FP)—the total number of heart sounds identified as abnormal but classified as normal.

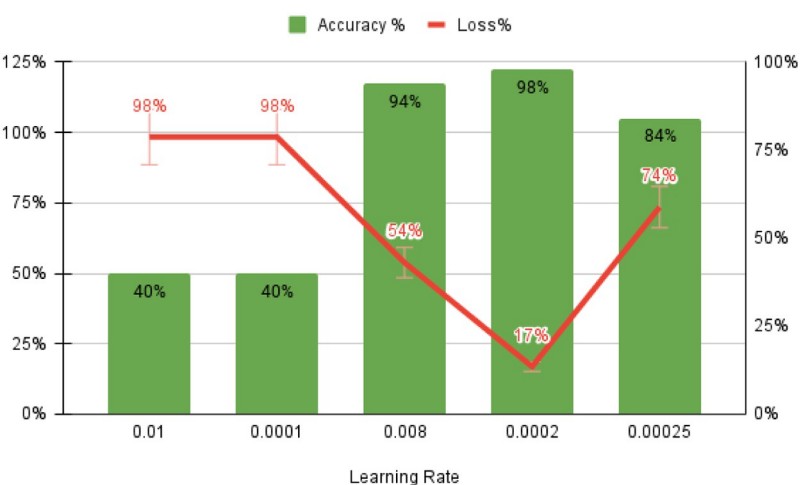

**Fig 13. Accuracy as a function of the rate of learning.**

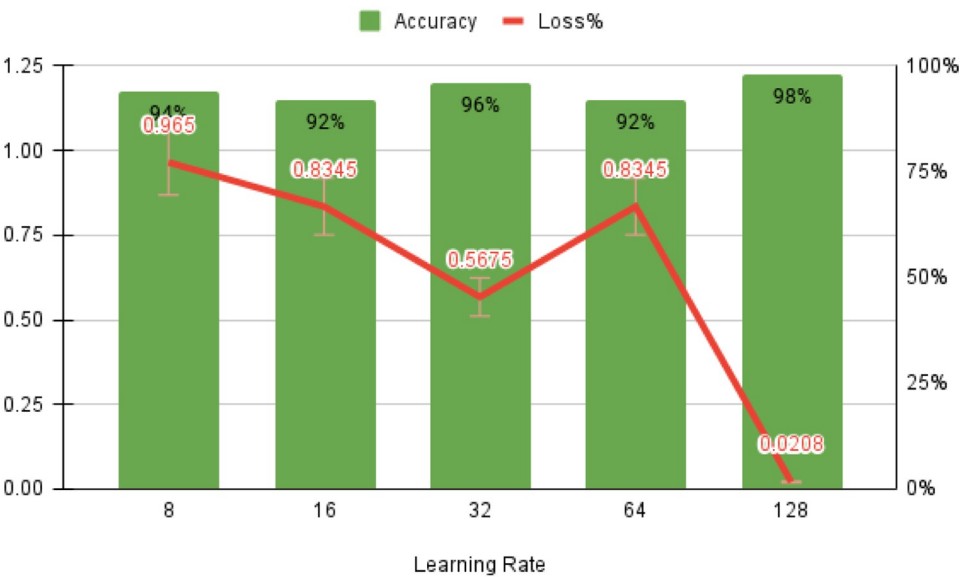

**Fig 14. Accuracy as the function of minibatch size.**

FN (False Negatives)—False Negatives (FN)—The total number of cardiac sounds that have been identified as normal and marked as pathological.

The proposed FFT + AlexNet performance evaluation and various classification methods are shown in Fig 15 and Table 2. The comparative performance analysis of the proposed model is shown in Figs 17 and 18. AlexNet's suggested transfer deep learning CNN approach achieves 99.7% accuracy, 98.3% sensitivity, 99.2% specificity, and 96.1% precision. The findings indicate that the proposed model outperforms other CNN algorithms regarding assessment measures. The WT with Feedforward neural network, FFT+ Multi-objective genetic algorithm, DFT with complex SVM, and WT with RF algorithm provided accuracy up to 88.2%, 98.70%, 98.25%, and 98.70%, respectively. Table 3 shows the comparison results of the proposed model and initial model based on Ranking based Average precision, F1-Score,

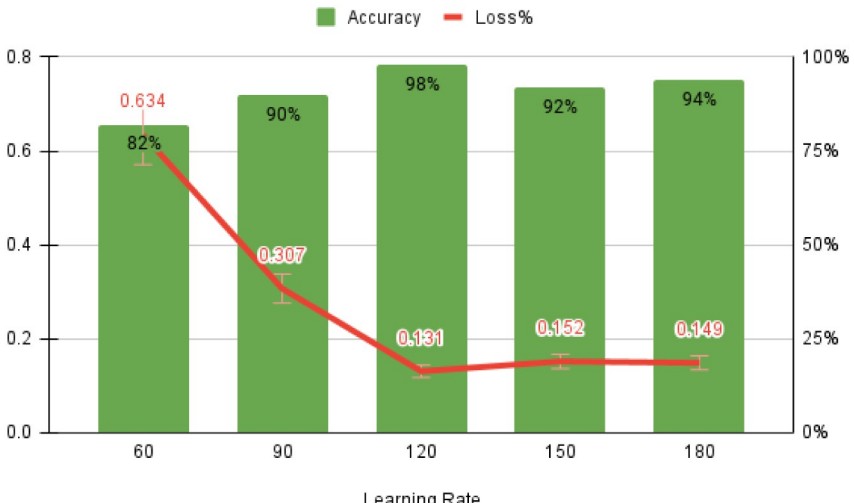

**Fig 15. Accuracy as the function of iteration.**

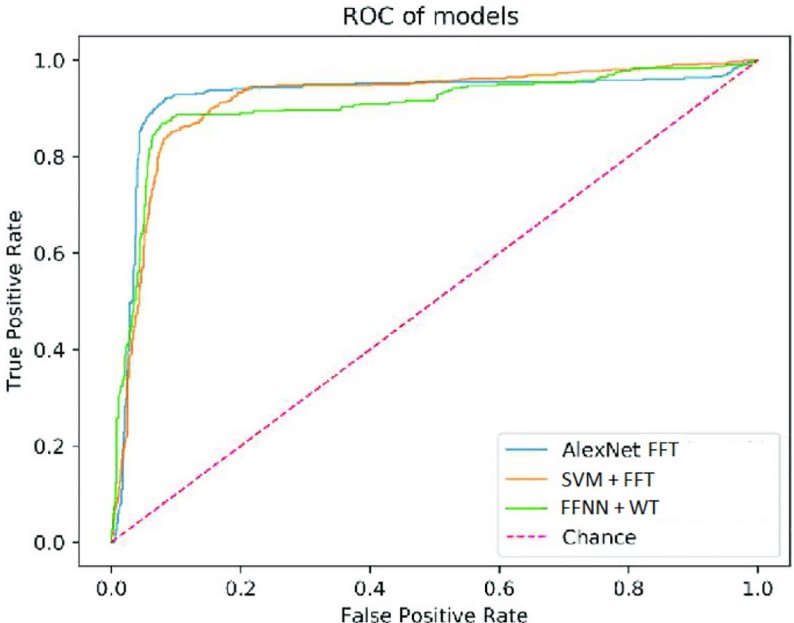

**Fig 16. Accuracy as the function of iteration.**

**Table 1. Collective result performance analysis and classification result using FFT and improved AlexNet.**

| Annotation | Classification performance result | | | |
|---|---|---|---|---|
| | **Sensitivity (ST)** | **Specificity (SP)** | **Positive predictive (PP)** | **Accuracy (ACC)** |
| NORMAL | 99.65 | 94.64 | 95 | 97 |
| PVC | 89.83 | 98.17 | 95 | 96.50 |
| APC | 91.50 | 98.58 | 97 | 97.17 |
| RBBB | 99.85 | 99.3 | 96 | 98.33 |

**Table 2. Comparative analysis of proposed works.**

| Feature Extraction method | Classification Method | Sensitivity (%) | Specificity (%) | Accuracy (%) | Precision (%) | References |
|---|---|---|---|---|---|---|
| WT | Feed forward neural network | 98 | 75 | 88.2 | 81.8 | [3] |
| FFT | Multi objective genetic algorithm | 97.5 | 98.3 | 98.70 | 95.4 | [4] |
| DFT | Complex SVM | 96.3 | 97.2 | 98.25 | 94.4 | [6] |
| WT | RF | 97.9 | 98.12 | 98.70 | 95.8 | [20] |
| FFT | **AlexNet** | **98.3** | **99.2** | **99.7** | **98.0** | **Proposed** |

Weighted Ranking Loss, and Coverage error. The results of the proposed FFT-based Improved ALEXNET algorithm performed better results based on F1-Score (98%) Coverage error (1.1189) and weighted ranking loss (0.0375). Compared to the initial model, the proposed algorithm produces better results.

## 5. Conclusion

The proposed biosignal ECG classification system shows that a probabilistic approach that combines improved AlexNet and Fast Fourier transform measurements provides better

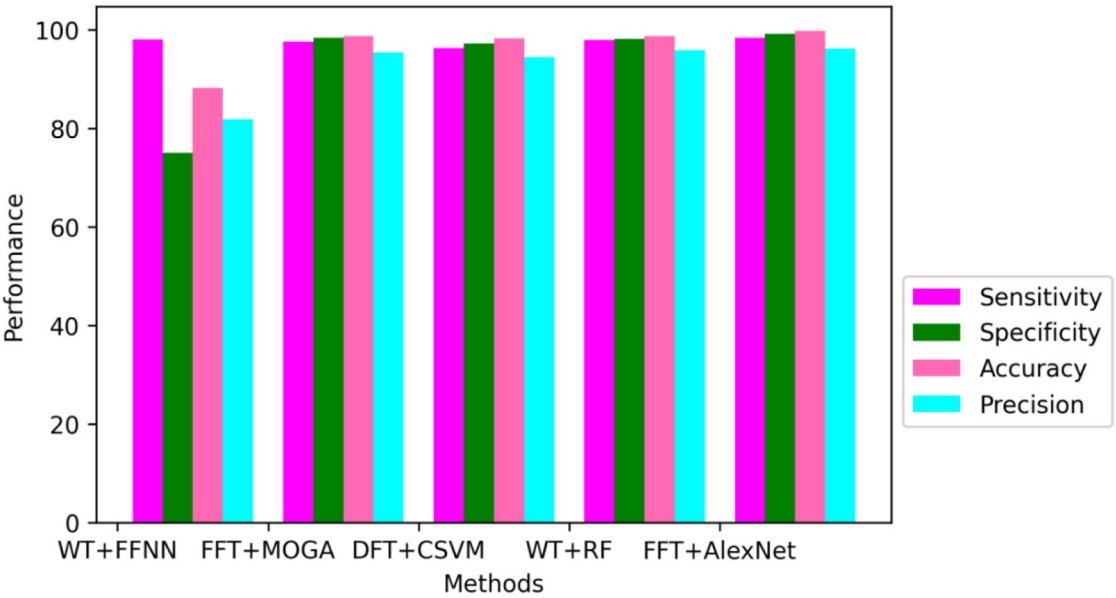

**Fig 17. The suggested model is compared to other ECG categorization algorithms.**

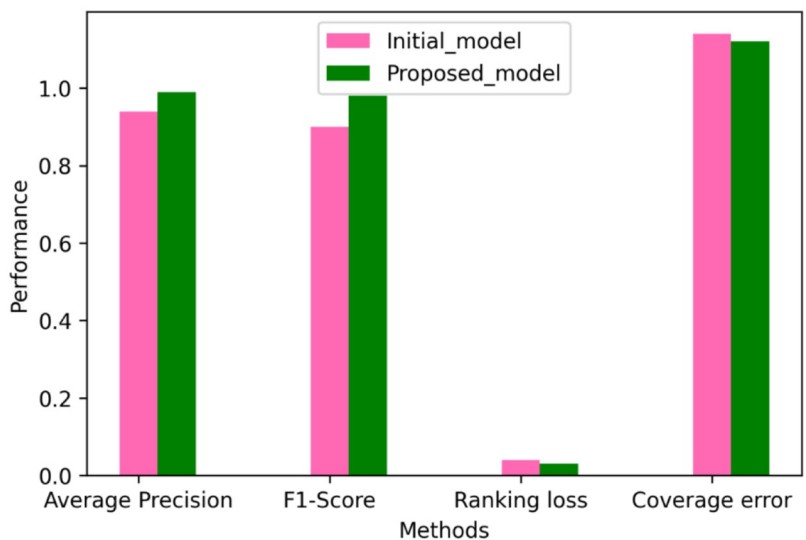

**Fig 18. Performance analysis of the proposed model.**

**Table 3. Results of the proposed model with the initial model.**

| Model | Ranking based on average precision | F1-Score | Weighted Ranking Loss | Coverage Error |
|---|---|---|---|---|
| Initial Model | 0.9486 | 0.90 | 0.0388 | 1.1374 |
| Proposed Model | 0.9943 | 0.98 | 0.0375 | 1.1189 |

recognition accuracy than conventional classifiers. Fast Fourier transform extracts a simplified set of functions from the input ECG signal and it is classified using an improved AlexNet classifier. The proposed technique using AlexNet attains 99.7% accuracy, 98.3% sensitivity, 99.2% specificity, and 96.1% precision. The results show that the proposed model is better than the conventional algorithms in terms of evaluation measures. The simulations are carried out in a variety of scenarios to verify the functionality of the proposed model. The experimental data prove that the proposed classifier outperforms WT with a Feedforward neural network, FFT with a Multi-objective genetic algorithm, DFT with complex SVM, and WT with an RF algorithm in terms of accuracy, specificity, sensitivity, and precision.

## Author Contributions

**Conceptualization:** Arun Kumar M.

**Data curation:** Arun Kumar M.

**Investigation:** Arun Kumar M.

**Methodology:** Arun Kumar M.

**Project administration:** Arvind Chakrapani.

**Supervision:** Arvind Chakrapani.

**Validation:** Arvind Chakrapani.

**Writing – original draft:** Arun Kumar M.

**Writing – review & editing:** Arvind Chakrapani.

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
