## [Decision Letter · Decision Letter 0]

27 Jun 2022

PONE-D-22-16995Classification of ECG Signal using FFT based Improved Alexnet ClassifierPLOS ONE

Dear Dr. M,

Thank you for submitting your manuscript to PLOS ONE. After careful consideration, we feel that it has merit but does not fully meet PLOS ONE’s publication criteria as it currently stands. Therefore, we invite you to submit a revised version of the manuscript that addresses the points raised during the review process.

We look forward to receiving your revised manuscript.

Kind regards,

Mohamed Hammad, Ph.D.

Academic Editor

PLOS ONE

Journal Requirements:

5. Please ensure that you refer to Figure 16 in your text as, if accepted, production will need this reference to link the reader to the figure.

Additional Editor Comments:

When updating your manuscript, you should elaborate on your points and clarify with references, examples, data, etc. Also, note that if a reviewer suggested references, you should only add those that are relevant to your work if you feel they strengthen your article.

Reviewers' comments:

Reviewer's Responses to Questions

**Comments to the Author**

1. Is the manuscript technically sound, and do the data support the conclusions?

Reviewer #1: No

Reviewer #2: Yes

Reviewer #3: No

2. Has the statistical analysis been performed appropriately and rigorously? 

Reviewer #1: No

Reviewer #2: Yes

Reviewer #3: No

3. Have the authors made all data underlying the findings in their manuscript fully available?

Reviewer #1: Yes

Reviewer #2: Yes

Reviewer #3: Yes

4. Is the manuscript presented in an intelligible fashion and written in standard English?

Reviewer #1: No

Reviewer #2: Yes

Reviewer #3: No

5. Review Comments to the Author

Reviewer #1: Manuscript is well written and formatted but less novelty in the work. Kindly follow AAMI standard in classification. The detailed review comments are attached in word file. Comparison of the proposed word with existing literature work needs to be included in the results section.

Reviewer #2: The authors performed good research onthe convolutional neural network technique to analyse ECG signal using improved FFT based AlexNet classifier. However, justification is required for the following comments,

1. Highlight the problem statement in the introduction section.

2. What are the merits of the proposed algorithm over traditional algorithms compared in the research work?

3. The related work section needs to be strengthened. I recommend the authors to discuss a few more recent research works.

4. In table 1, what are PVC, APC, and RBBB?

5. What is the modification done in AlexNet architecture for stating it as improved AlexNet?

6. The introduction is too long, please remove the repeated idea. No need to explain much in detail. More explanation should be about the novelty of the proposed method.

7. In section 3.4, FFT is used for feature extraction. How the extraction was done and what are the extracted features?Include the expression for the same corresponding to the proposed architecture.

Reviewer #3: Summary: The authors propose an improved AlexNet based on FFT. Results seem to back the technique.

My comments:

1. The manuscript in its current form is ambiguous, confusing and unclear.

2. Narrations are incoherent.

3. Many sentences are presented as if they were section headings.

4. There are repetitions of the same sentences throughout the manuscript.

5. Figures are barely visible and sometimes (Figure 4) tend not to depict what the authors are stating in the text.

6. Some figures tend to explain things better than how the authors narrate things, which again goes towards the disadvantage of the authors.

7. In the manuscript, authors write “The AlexNet architecture is depicted in Figure 5” and then the caption of Figure 5 reads, “Architecture of the improved AlexNet model”. I wonder whether Figure 5 is original AlexNet or the authors’ improved version.

8. Acronyms have not been properly introduced.

9. Algorithms/pseudo-codes have not been presented wherever needed.

10. Data distributions/dataset analyses have not been provided.

11. Authors claim to propose AlexNet based on FFT in the Abstract, which is wrong. AlexNet is already a CNN and the authors are proposing to use FFT for feature extraction of ECG signals, and later use AlexNet for classification. Or rather, they are improving AlexNet architecture as the authors claim. Again, the language needs to be sorted out here.

12. Then, authors write in Section 3.4, Paragraph 2, “Fast Fourier transform technology has successfully extracted feature components from ECG data, such as PQRST signals”. If FFT has done it successfully, what exactly are the authors accomplishing in the current manuscript?

13. The authors write in Section 3.5, last Paragraph, “Each ECG image is transmitted to the improved AlexNet in this stage”. Which stage are the authors referring to here? Again, one can see the issue with the language.

My conclusion: My main problem with this manuscript is the way it has been presented. The language is all over the place and is found to be naïve at many places as well. The results might seem to be good, however, since the paper is hard to understand and follow, they might not account to much. This, unfortunately, does not qualify the standards of Plos One. The authors need to reorganise the manuscript, revisit the language and seek professional assistance to improve on the above-mentioned shortcomings. Having said that, I believe the idea and the technique presented in the manuscript are intriguing and it appears the results back the technique. Therefore, the authors should revisit the whole manuscript and overcome the shortcomings and resubmit to Plos One. However, as things stand at the moment, my decision is of rejection.

6. PLOS authors have the option to publish the peer review history of their article (what does this mean?). If published, this will include your full peer review and any attached files.

Reviewer #1: **Yes: **Allam Jaya Prakash

Reviewer #2: **Yes: **Anandakumar H

Reviewer #3: No

---

## [Author Response · Author response to Decision Letter 0]

2 Aug 2022

Reviewer #1:

1. Kindly write the abstract in a concise and succinct manner, there is no requirement of background basic study in the abstract. The first three to four lines can shift to the introduction.

Based on the suggestion, the abstract is revised and a few lines were shifted to the introduction section. The following modification is carried out and included in the revised manuscript.

Cardiovascular disease is currently a severe hazard to life and health, and its incidence is increasing year after year. As a result, it is critical to concentrate on cardiovascular disease diagnosis and prevention. An electrocardiogram (ECG) is the most popular non-invasive screening test to identify a specific cardiac issue. It keeps track of the heart's function throughout time.

2. Use convolutional neural network short term in abstract.

In the abstract, Convolutional Neural Network is replaced by “CNN”

3. Why authors are concentrated on only four beats, any specific reason?

Right Bundle Branch Block (RBBB), Premature Atrial Contraction (PAC), and Premature Ventricular Contraction (PVC) are some of the significant heartbeat problems investigated in the research paper. These four beats are the major types for ECG analysis.

4. Better to use any customized network not transfer learning 

Thanks to the reviewer for the suggestion. In this research, improved AlexNet along with FFT is used to process ECG signals. Transfer learning can also be adapted to process EEG signals rather than ECG signals using the same network architecture.

5. Figure 2 is copied from the internet

Figure 2 is a common picture of the ECG signal used to represent the P, Q, R, and S factors. 

6. I didn’t find any comparison of the proposed method with existing literature 

In the literature section, the following statements were introduced as per the suggestion.

There are numerous accounts of ECG beat classification using a variety of techniques, including artificial neural networks (ANN), self-organizing maps (SOM), support vector machine (SVM) classifiers, soft independent modeling of class analogy (SIMCA), deep learning, and complex support vector machines (CSVM), decision trees, and convolution neural network (CNN).

SVM exhibits poor behavior in the face of class instabilities, but methods of handling have been developed, including hierarchical SVM or SVM weighted by each class. The drawback is that, in very large problems, it may not always be possible to find an optimization, and the training algorithm is not guaranteed to achieve a global optimization.

Due to the high computational cost during the test phase, the KNN-k nearest neighbor method (KNN) has limited application in real-world scenarios. Decision Tree is not commonly used because it can only handle a limited number of features and the rule-based approach performs the worst. 

The results of the existing method are compared with the proposed method.

7. Write specific author contribution

Major contribution: 

• Numerous algorithms have been used in research, including random forests and decision tree ensembles as well as the non-linear classifier Support Vector Machine (SVM). Manual feature extraction is necessary for this algorithm. In this research neural networks are used to address the issue in medical diagnostics. This increases the effectiveness of training techniques, makes it possible for the algorithm to be used "end-to-end," and makes it simpler for a wider range of people if large datasets are available.

8. MIT-BIH full form missed

The full form of MIT-BIH is Massachusetts Institute of Technology-Beth Israel Hospital. In the revised manuscript, the full form is provided.

9. Please remove unnecessary capitalizations in the manuscript

Unnecessary capitalizations are removed in the revised manuscript. 

10. Please mention important major contributions only

The goal of this research is to develop a deep learning-based method to detect arrhythmias automatically without the use of manual feature identification.

11. Quality of the images are very poor, and not visible also

The quality of the images is enhanced in the revised manuscript.

12. As per my knowledge authors are used Python for implementation, but network diagram missed?

Instead of the network diagram, the visualization of the AlexNet is provided in Figure 5.

13. Needs to include ROC curves, and precision-recall curves

The above ROC curve plot is included in the revised manuscript.

14. Grammatically needs to recheck again

The revised manuscript is checked with Grammar correction using a grammar correcting tool.

15. Needs to recheck the structure of the paper as Introduction, Literature, and Motivation, Contributions of the proposed work, Database, Proposed methodology, Experimental results, Discussion, Conclusion + Future scope.

The structure of the paper is reframed as per the comments.

16. Needs to include ablation study (If possible)

Ablation was carried out wherever possible

17. Conclusions also needs to re-write again

The conclusion section is rewritten again with appropriateness and the future work is also included.

18. Please cite recent scripts as below

The references recommended by the reviewer are also included in the revised manuscript as follows,

• Prakash, A. J., Samantray, S., Bala, C. L., & Narayana, Y. V. (2021). An Automated Diagnosis System for Cardiac Arrhythmia Classification. In Analysis of Medical Modalities for Improved Diagnosis in Modern Healthcare (pp. 301-313). CRC Press.

• Prakash, A. J., & Ari, S. (2019, December). AAMI standard cardiac arrhythmia detection with random forest using mixed features. In 2019 IEEE 16th India Council International Conference (INDICON) (pp. 1-4). IEEE.

• Allam, J. P., Samantray, S., & Ari, S. (2020). SpEC: A system for patient specific ECG beat classification using deep residual network. Biocybernetics and Biomedical Engineering, 40(4), 1446-1457.

• Prakash, A. J., & Ari, S. (2019). A system for automatic cardiac arrhythmia recognition using electrocardiogram signal. In Bioelectronics and Medical Devices (pp. 891-911). Woodhead Publishing.

• Hammad, M., Pławiak, P., Wang, K., & Acharya, U. R. (2021). ResNet‐Attention model for human authentication using ECG signals. Expert Systems, 38(6), e12547.

• Tuncer, T., Dogan, S., Pławiak, P., & Acharya, U. R. (2019). Automated arrhythmia detection using novel hexadecimal local pattern and multilevel wavelet transform with ECG signals. Knowledge-Based Systems, 186, 104923.

• Książek, W., Gandor, M., & Pławiak, P. (2021). Comparison of various approaches to combine logistic regression with genetic algorithms in survival prediction of hepatocellular carcinoma. Computers in Biology and Medicine, 134, 104431.

Reviewer #2: 

The authors performed good research on the convolutional neural network technique to analyze ECG signals using improved FFT-based AlexNet classifier. However, justification is required for the following comments,

1. Highlight the problem statement in the introduction section.

The problem statement is highlighted in the introduction section.

2. What are the merits of the proposed algorithm over traditional algorithms compared in the research work?

Random forests and decision tree ensembles as well as the non-linear classifier Support Vector Machine (SVM) are compared in this research along with the proposed classifier. Manual feature extraction is necessary for this algorithm. In this research neural networks are used to address the issue in medical diagnostics. This increases the effectiveness of training techniques, makes it possible for the algorithm to be used "end-to-end," and makes it simpler for a wider range of people if large datasets are available.

3. The related work section needs to be strengthened. I recommend the authors discuss a few more recent research works.

Recent literature related to the research work is included and cited.

4. In table 1, what are PVC, APC, and RBBB?

Right Bundle Branch Block (RBBB), Premature Atrial Contraction (PAC), and Premature Ventricular Contraction (PVC) are some of the significant heartbeat problems investigated in this research.

5. What is the modification done in AlexNet architecture for stating it as improved AlexNet?

Thanks to the reviewer for the suggestion. The following characteristics distinguish the improvements proposed in the paper from the traditional AlexNet network classification algorithm:

• An additional convolution layer is introduced to the original AlexNet structure and the max-avg pooling technique is used to preserve the local receptive fields. This will provide more accurate image feature information.

• The global average pooling (GAP) layer is substituted for the original fully connected FC layer, which significantly reduces the over-fitting effect without affecting the final features. The final result is unaffected in the absence of numerous calculations of network parameters, increasing network speed.

• The LRN layer is added to the convolution layer to avoid some unnecessary numerical issues; this effectively avoids neuron saturation. The BN layer in the proposed method is used after the convolution of each layer.

6. The introduction is too long, please remove the repeated idea. No need to explain much in detail. More explanation should be about the novelty of the proposed method.

The introduction section is reduced and the novelty of the proposed methodology is briefed.

7. In section 3.4, FFT is used for feature extraction. How the extraction was done and what are the extracted features? Include the expression for the same corresponding to the proposed architecture.

Based on the suggestion, section 3.4 is revised. In this study, the entire segment associated with the Q, R, and S-peak, and through these deflections QRS complex and RR interval wave was identified and included in the revised manuscript.

Reviewer #3

1. The manuscript in its current form is ambiguous, confusing and unclear.

The manuscript is restructured clearly.

2. Narrations are incoherent.

The narrations represented in the manuscript are updated.

3. Many sentences are presented as if they were section headings.

All the sentences are reframed and introduced in the subsections.

4. There are repetitions of the same sentences throughout the manuscript.

Repetitions are corrected in the revised manuscript. 

5. Figures are barely visible and sometimes (Figure 4) tend not to depict what the authors are stating in the text.

The quality of the images is enhanced for better visibility.

6. Some figures tend to explain things better than how the authors narrate things, which again goes towards the disadvantage of the authors.

The description of the figures is included in the revised manuscript.

7. In the manuscript, the authors write “The AlexNet architecture is depicted in Figure 5” and then the caption of Figure 5 reads, “Architecture of the improved AlexNet model”. I wonder whether Figure 5 is the original AlexNet or the authors’ improved version.

The AlexNet network structure is given in Figure 5. The improved AlexNet network structure is given in Figure 6.

8. Acronyms have not been properly introduced.

The entire manuscript is rechecked and the acronyms are properly mentioned.

9. Data distributions/dataset analyses have not been provided.

Data distributions are highlighted in the dataset section 3.2.

10. Authors claim to propose AlexNet based on FFT in the Abstract, which is wrong. AlexNet is already a CNN and the authors are proposing to use FFT for feature extraction of ECG signals, and later use AlexNet for classification. Or rather, they are improving AlexNet architecture as the authors claim. Again, the language needs to be sorted out here.

The abstract is rewritten with correctness. This paper proposes the use of an ECG arrhythmia classification scheme based on Fast Fourier Transform (FFT) for feature extraction and an improved AlexNet classifier to differentiate between four types of arrhythmias conditions that were obtained from records.

11. Then, the authors write in Section 3.4, Paragraph 2, “Fast Fourier transform technology has successfully extracted feature components from ECG data, such as PQRST signals”. If FFT has done it successfully, what exactly are the authors accomplishing in the current manuscript?

Each complex ECG signal has a real and an imaginary component. The Fast Fourier Transform removes low frequencies from the ECG signal. The inverse fast Fourier transform is used to remove noise. The Fast Fourier Transform is used to transform the input signal from the dataset after it has been preprocessed by removing nulls. The extracted features were suitable for detecting arrhythmia in patient records, and the results were obtained using Matlab. The created Excel file can then be used to classify and detect various anomalies.

12. The authors write in Section 3.5, last Paragraph, “Each ECG image is transmitted to the improved AlexNet in this stage”. Which stage are the authors referring to here? Again, one can see the issue with the language. 

The description of the stage is included in the revised manuscript. Each ECG image is transmitted to the improved AlexNet in this classification stage. The individual ECG beats that were recovered from the Fourier coefficients were further divided into two groups for training and testing to facilitate classification. Using the FFT coefficient as the function vector in the classifier's input vector, the improved AlexNet classifier is used to differentiate between the four different types of ECG arrhythmias.

---

## [Decision Letter · Decision Letter 1]

8 Aug 2022

PONE-D-22-16995R1Classification of ECG Signal using FFT based Improved Alexnet ClassifierPLOS ONE

Dear Dr. M,

Thank you for submitting your manuscript to PLOS ONE. After careful consideration, we feel that it has merit but does not fully meet PLOS ONE’s publication criteria as it currently stands. Therefore, we invite you to submit a revised version of the manuscript that addresses the points raised during the review process.

We look forward to receiving your revised manuscript.

Kind regards,

Mohamed Hammad, Ph.D.

Academic Editor

PLOS ONE

Journal Requirements:

Reviewers' comments:

Reviewer's Responses to Questions

**Comments to the Author**

1. If the authors have adequately addressed your comments raised in a previous round of review and you feel that this manuscript is now acceptable for publication, you may indicate that here to bypass the “Comments to the Author” section, enter your conflict of interest statement in the “Confidential to Editor” section, and submit your "Accept" recommendation.

Reviewer #1: All comments have been addressed

Reviewer #2: All comments have been addressed

2. Is the manuscript technically sound, and do the data support the conclusions?

Reviewer #1: Yes

Reviewer #2: Yes

3. Has the statistical analysis been performed appropriately and rigorously? 

Reviewer #1: Yes

Reviewer #2: Yes

4. Have the authors made all data underlying the findings in their manuscript fully available?

Reviewer #1: Yes

Reviewer #2: Yes

5. Is the manuscript presented in an intelligible fashion and written in standard English?

Reviewer #1: Yes

Reviewer #2: Yes

6. Review Comments to the Author

Reviewer #1: Authors are incorporated all my comments. Only two minor suggestions from my side

1. Figure 2 should be changed; the author can generate one ECG beat segment using MATLAB or Python. The copied figure from the internet should be replaced with a new figure before publication.

Reviewer #2: The author has addressed all the necessary comments and the revised manuscript is improved .

Recommended for further publications

7. PLOS authors have the option to publish the peer review history of their article (what does this mean?). If published, this will include your full peer review and any attached files.

Reviewer #1: **Yes: **Allam Jaya Prakash

Reviewer #2: **Yes: **Anandakumar H

---

## [Author Response · Author response to Decision Letter 1]

14 Aug 2022

Reviewer #1:

Authors are incorporated all my comments. Only two minor suggestions from my side.

Comment 1: Figure 2 should be changed; the author can generate one ECG beat segment using MATLAB or Python. 

Response: Thank you for valuable suggestion. As per the comment, one ECG beat segment is generated using python and the same is included in the revised manuscript.

Comment 2: The copied figure from the internet should be replaced with a new figure before publication.

Response: Figure 2 is replaced with the new one in the revised paper.

---

## [Editor Report · Decision Letter 2]

24 Aug 2022

Classification of ECG Signal using FFT based Improved Alexnet Classifier

PONE-D-22-16995R2

Dear Dr. M,

We’re pleased to inform you that your manuscript has been judged scientifically suitable for publication and will be formally accepted for publication once it meets all outstanding technical requirements.

Kind regards,

Mohamed Hammad, Ph.D.

Academic Editor

PLOS ONE
---

## [Editor Report · Acceptance letter]

7 Sep 2022

PONE-D-22-16995R2 

Classification of ECG Signal using FFT based Improved Alexnet Classifier 

Dear Dr. Kumar M.:

I'm pleased to inform you that your manuscript has been deemed suitable for publication in PLOS ONE. Congratulations! Your manuscript is now with our production department. 

Kind regards, 

on behalf of

Dr. Mohamed Hammad 

Academic Editor

PLOS ONE